# Echoes of the Visual Past: Test-Time Prompt Tuning with Multi-Scale Visual Memory

## Abstract

Test-time prompt tuning (TPT) aims to adapt pre-trained vision-language models (VLMs) to various downstream tasks by learning textual prompts using unlabeled data at test time. However, existing TPT methods exhibit a performance gap compared to a line of prompt-engineering-based methods that leverage hand-crafted or LLM-generated prompts for VLM adaptation. We attribute this gap to a core limitation of previous TPT approaches: they learn prompts from only limited class-specific visual knowledge derived from a single test image. As a result, the learned prompts underperform compared to hand-crafted and LLM-generated prompts enriched with diverse, class-specific knowledge. To address this limitation, we propose **T**est-time **P**rompt **T**uning with **M**ulti-scale visual **M**emory ($M^2TPT$). Specifically, the memory is constructed to store past seen class-relevant image patches as multi-scale visual descriptions for each class. For each test image, we use it to query the memory and learn the textual prompt using both the test image and the retrieved class-relevant visual memory. Additionally, we introduce holistic visual memory to better handle holistic visual recognition tasks that require global image-level context, and an irrelevance suppression strategy to mitigate the impact of noisy memory entries at test time. We evaluate our method on 15 commonly used benchmark datasets and show that it outperforms existing TPT methods. Furthermore, our framework can incorporate human-designed prompts and achieves state-of-the-art performance compared to recent VLM adaptation methods that use hand-crafted or LLM-generated prompts.

## 1 Introduction

Pre-trained vision-language models (VLMs) have demonstrated powerful representational capabilities, making them valuable for a wide range of computer vision tasks (Radford et al., 2021; Jia et al., 2021; Li et al., 2022; 2023; Liu et al., 2023). To efficiently adapt VLMs to downstream tasks and new domains, the prompt-tuning paradigm has been explored—where only the input text context is optimized using limited test data, while the model backbone remains frozen (Zhou et al., 2022b;a). More practically, recent research has developed test-time prompt tuning (TPT), which directly optimizes prompts using unlabeled test data streams (Shu et al., 2022).

Aside from prompt-tuning-based methods, a line of prompt-engineering-based methods have designed hand-crafted and LLM-generated prompts tailored for each dataset to adapt VLMs to target tasks (Pratt et al., 2023; Karmanov et al., 2024; Zhang et al., 2024d; Zhu et al., 2024). Recently these methods have significantly outperformed prompt-tuning approaches on image classification benchmarks, as illustrated in Fig. 1a. Human-designed prompts introduce prior dataset knowledge and rich class-specific information, making them more effective than prompts learned from a generic "a photo of a [CLASS]" initialization during test time. As the red dashed lines show the performance of these methods when using a generic prompt, we found that the performance gap mainly lies between the *learned prompt* and the *human-designed prompts*. However, prompt-engineering-based methods require prior knowledge of the test datasets and additional time or effort to design or generate effective prompts. In contrast, TPT methods can adapt to unlabeled test streams on the fly without relying on human intervention. Has the potential of TPT methods truly been exhausted?

As depicted in Fig. 1b, prior TPT methods typically optimize a trainable prompt using only the current test image and its augmentations, relying on unsupervised losses such as entropy minimiza-

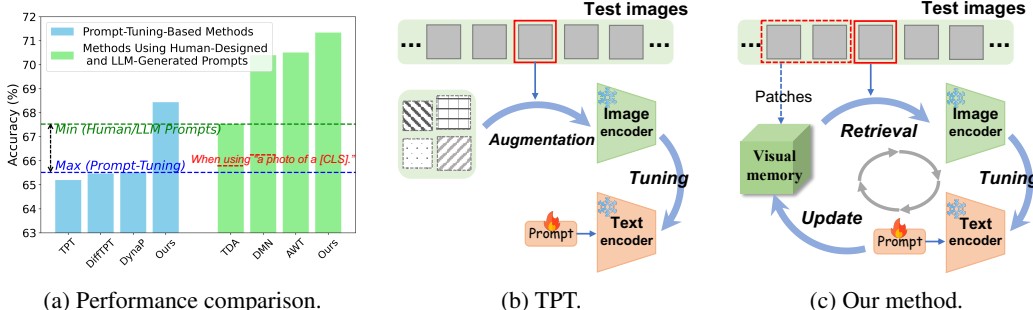

(a) Performance comparison.       (b) TPT.       (c) Our method.

Figure 1: **(a) Performance comparison on 10 downstream classification datasets.** Existing test-time prompt tuning (TPT) methods exhibit a performance gap compared to prompt-engineering-based methods, as illustrated by the blue and green dashed lines. The red dashed lines highlight that the performance gap primarily lies between the TPT-learned prompt and the human-designed prompts. **(b) TPT** (Shu et al., 2022). Previous TPT methods typically optimize a learnable prompt using only the visual information from the current test image and its augmentations. **(c) Our method** enhances test-time prompt learning with memorized past visual descriptions for each class and introduces a mutual promotion framework between the learnable prompt and the evolving visual memory.

tion (Shu et al., 2022) and distribution alignment (Abdul Samadh et al., 2023). We argue that such methods fail to learn prompts from sufficient class-specific visual knowledge due to their reliance on limited visual information from a current test image, which limits their competitiveness compared to human-designed prompts enriched with diverse and explicit class- and dataset-level knowledge. To address this limitation, we propose test-time prompt tuning enhanced by a past visual memory containing class-specific visual descriptions.

Our approach, as depicted in Fig. 1c, constructs a multi-scale visual memory by accumulating visual patches that are highly relevant to each class at every time step during the test stream. Before prompt tuning, the current test image is used as a query to retrieve semantically related visual patches from this memory. The textual prompt is then optimized using both the test image and the retrieved, diverse, class-relevant visual information from the same test distribution, enabling the prompt tuning process to more effectively capture class-specific knowledge. Reciprocally, the visual memory also benefits from the learned prompt, as it is updated based on the optimized prompt. The three sequential steps—memory retrieval, prompt tuning, and memory update—achieve a round of mutual promotion between the tunable textual prompt and the evolving visual memory for each test image. This results in a dual-branch prediction mechanism where the final output is derived jointly from the tuned prompt and the evolving visual memory.

In addition to object recognition, downstream tasks may require holistic visual understanding, such as scene understanding (Xiao et al., 2010) and land cover classification (Helber et al., 2019), which demand comprehensive image-level context that may be lost when focusing solely on patches. To this end, we further construct a holistic visual memory that retains class-relevant full-view images and functions in coordination with the multi-scale memory. Moreover, because memory update and retrieval operate without ground-truth supervision at test time, the visual memory can inevitably be noisy due to prediction errors. To mitigate adverse effects, we introduce an irrelevance suppression strategy: we filter out low-relevance memory entries from the retrieved class-specific memory during retrieval, and we maintain a class-irrelevant memory that stores previously seen misleading patches from the test domain. This irrelevant memory is used to penalize high-confidence but incorrect cues during prompt tuning, thereby suppressing distracting and misleading information.

We evaluate our test-time prompt tuning method on 15 datasets, including commonly used downstream image classification benchmarks and out-of-distribution datasets. Our method outperforms existing test-time prompt tuning methods without prompt engineering. Furthermore, our framework can also benefit from human-designed prompts, enabling it to achieve state-of-the-art performance compared to recent VLM adaptation methods that rely on hand-crafted or LLM-generated prompts.

## 2 RELATED WORK

**Prompt learning.** As vision-language models (VLMs) have demonstrated strong performance across various computer vision tasks, recent research has explored prompt learning as a parameter-efficient approach to adapt VLMs to real-world downstream scenarios (Lu et al., 2022; Hantao Yao, 2023; Bulat & Tzimiropoulos, 2023; Lee et al., 2023; Kan et al., 2023; Khattak et al., 2023b). CoOp (Zhou et al., 2022b) proposes learning a contextual prompt in the input space of the text encoder using few-shot data, while keeping the model backbone frozen. CoCoOp (Zhou et al., 2022a) improves upon CoOp by introducing condition tokens derived from input images into the textual prompt learning process, enabling better generalization. In contrast, Bahng et al. (Bahng et al., 2022) introduce visual prompt learning, which operates on the image encoder of VLMs. MaPLe (Khattak et al., 2023a) further advances this line of work by jointly learning prompts on both the image and text encoders to enhance transfer learning performance.

**Test-time prompt tuning.** To improve the generalization ability of VLMs without requiring labeled test data, TPT (Shu et al., 2022) proposes test-time prompt tuning (TPT). This pioneering method learns adaptive textual prompts from the current test image and its augmentations using an entropy minimization objective, while keeping the model backbone frozen. PromptAlign (Abdul Samadh et al., 2023) explicitly addresses distribution shift by introducing a distribution statistics alignment loss to guide test-time prompt optimization. C-TPT (Yoon et al., 2024) considers the calibration of VLMs for prompt tuning at test time. More recently, HisTPT (Zhang et al., 2024b) and DynaPrompt (Xiao et al., 2025) propose online TPT methods to leverage past information during inference. HisTPT (Zhang et al., 2024b) constructs long-term and short-term knowledge banks that store output text features generated from prompts, providing self-regularization to stabilize online prompt learning. DynaPrompt (Xiao et al., 2025) maintains a prompt pool containing multiple prompts and selects among them for stable online optimization. In our method, we do not follow this continuous TPT paradigm, but instead adopt the original setting introduced by TPT (Shu et al., 2022), in which an adaptive prompt is learned from scratch for each test sample independently.

**VLM adaptation with prompt engineering.** Apart from prompt-tuning-based methods, another line of research explores prompt-engineering-based VLM adaptation (Menon & Vondrick, 2022; Roth et al., 2023; Pratt et al., 2023; Novack et al., 2023; Ge et al., 2023). CuPL (Pratt et al., 2023) leverages large language models (LLMs) (Achiam et al., 2023) to generate textual descriptions for each class in the test dataset, replacing the generic prompt with these customized ones to improve prediction accuracy. TDA (Karmanov et al., 2024) and DMN (Zhang et al., 2024d) adopt hand-crafted prompts and LLM-generated prompts, respectively, on the text branch. On the vision branch, they design memory-based methods that perform non-parametric learning with visual features in a manner similar to the k-nearest neighbors (KNN) algorithm (Mucherino et al., 2009), to improve zero-shot classification. BoostAdapter (Zhang et al., 2024c) further introduces boosting samples, i.e., augmented multi-scale views of the current test image, and combines them with the historical memory at each test step. Besides, DPE (Zhang et al., 2024a) and GS-Bias (Huang et al., 2025) employ prototype and bias learning behind the encoders, leaving the textual prompt free for prompt engineering. MCP (Chen et al., 2025) proposes a multi-cache enhanced prototype learning framework that incorporates both a memory-based mechanism and a prototype learning component. Moreover, AWT (Zhu et al., 2024) uses LLMs to generate class-specific prompt candidates and transforms the test image into multiple views, then formulates image–text matching as an optimal transport problem for zero-shot classification. While prompt-engineering-based methods have demonstrated effectiveness, they require prior knowledge of the test dataset and additional effort to craft or generate prompts. In contrast, TPT methods aim to adapt VLMs on the fly, focusing on test-time prompt learning without relying on human supervision.

## 3 METHOD

In this section, we first introduce the preliminaries of CLIP and test-time prompt tuning in Sec. 3.1. Then, Sec. 3.2 describes the overall framework of our method and its main component, the multi-scale visual memory. Secs. 3.3 and 3.4 present the additional two components of our methods.

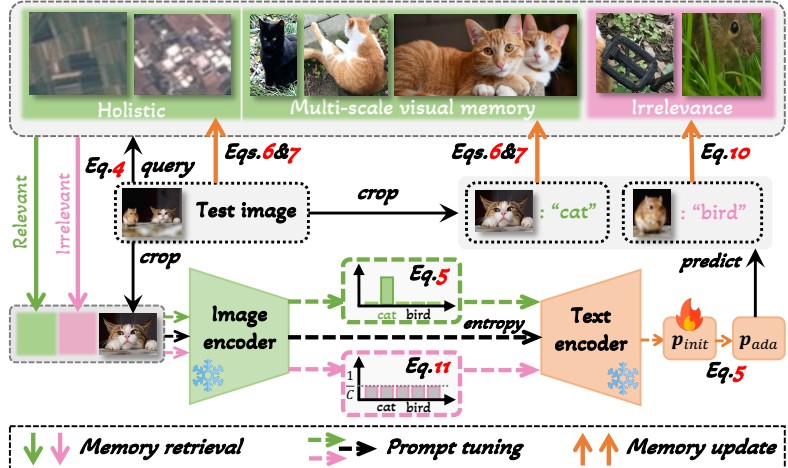

Figure 2: Overview of our method. Each test image undergoes three sequential steps: memory retrieval, prompt tuning, and memory update. In the memory retrieval step, the test image is used to query both the multi-scale memory and the holistic memory. The predicted label is then used to fetch class-relevant patches or images, as well as misleading patches from the class-irrelevant memory. During prompt tuning, the textual prompt is optimized using the test image and the retrieved visual memory. Finally, in the memory update step, the adapted prompt is used to update the class-relevant patches and the test image in the multi-scale and holistic memories, while high-confidence but irrelevant patches are added to the class-irrelevant memory.

## 3.1 PRELIMINARIES

**CLIP.** The Contrastive Language–Image Pre-training (CLIP) model (Radford et al., 2021) comprises an image encoder $f(\cdot)$ and a text encoder $g(\cdot)$, which are pre-trained on large-scale image–text pairs using contrastive learning. Once pre-trained, CLIP can perform zero-shot image classification on a variety of downstream datasets. For a dataset with $C$ classes, CLIP first encodes each class using a generic prompt $\mathbf{p}$, such as "a photo of [CLASS c].", producing class-specific text embeddings $\{\mathbf{p}_c\}_{c=1}^{C}$. Given a test image $\mathbf{X}$, CLIP compares its encoded feature $f(\mathbf{X})$ with the text embeddings of all classes and computes the probability of class membership as follows:

$$p(y = c \mid \mathbf{X}, \mathbf{p}) = \frac{\exp(\cos(f(\mathbf{X}), g(\mathbf{p}_c))/\tau)}{\sum_{j=1}^{C} \exp(\cos(f(\mathbf{X}), g(\mathbf{p}_j))/\tau)}, \tag{1}$$

where $\cos(\cdot, \cdot)$ denotes cosine similarity, and $\tau$ is a learned temperature parameter. For clarity, we assume that the outputs of the encoders are normalized by default throughout the rest of the paper.

**Test-time prompt tuning.** Directly applying CLIP to downstream tasks may suffer from performance degradation due to distribution shifts. Test-time prompt tuning (TPT) aims to adapt CLIP to the test data by optimizing a prompt at test time. For instance, the pioneering TPT method (Shu et al., 2022) augments the test image $\mathbf{X}$ into $N$ views $\mathbf{X}_{[N]}$, and then optimizes a prompt $\mathbf{p}$ using $n$ selected augmentations $\mathbf{X}_{[n]}$ with low entropy, based on an entropy minimization loss:

$$\mathcal{L} = -\sum_{c=1}^{C} \bar{p}(y = c \mid \mathbf{X}_{[n]}, \mathbf{p}) \cdot \log \bar{p}(y = c \mid \mathbf{X}_{[n]}, \mathbf{p}), \tag{2}$$

where $\bar{p}(y = c \mid \mathbf{X}_{[n]}, \mathbf{p})$ denotes the average prediction probability across augmentations $\mathbf{X}_{[n]}$.

## 3.2 MULTI-SCALE VISUAL MEMORY

Previous TPT methods typically use only the current test image or its augmentations to learn a tunable prompt. However, the visual class information available from a single test image is limited for prompt learning. As a result, TPT methods significantly underperform recent human-designed prompt methods, as shown in Fig. 1a. To address this limitation, we propose TPT enhanced by multi-scale visual memory, which provides diverse visual class information from past data to guide

prompt learning. Specifically, our method integrates visual memory into the test-time prompt tuning workflow and introduces a **prompt-memory mutual promotion framework**. As illustrated in Fig. 2, for each test sample, the method involves three sequential steps: **Memory retrieval**, **Prompt tuning**, and **Memory update**.

**Memory retrieval.** Let the multi-scale visual memory be denoted as $\mathcal{M} \in \mathbb{R}^{C \times S \times D}$, where $C$ is the number of classes, $S$ is the memory size per class, and $D$ is the dimension of image patches. For a given test image $\mathbf{X} \in \mathbb{R}^{3 \times H \times W}$, we calculate the similarity between the memorized patches and the image in the feature space encoded by the CLIP image encoder. Specifically, we denote the encoded test image as $f(\mathbf{X}) = \mathbf{v} \in \mathbb{R}^d$, and the encoded memory as $f(\mathcal{M}) = \mathbf{M} \in \mathbb{R}^{C \times S \times d}$. We define the $(c, m)$-th memory vector as $\mathbf{m}_{c,m} := \mathbf{M}[c, m] \in \mathbb{R}^d$. The similarity is computed as:

$$\mathbf{S}_{c,m} = \phi\left(\mathbf{v}^\top \mathbf{m}_{c,m}\right), \quad \text{for } c = 1, \ldots, C, \ m = 1, \ldots, S, \tag{3}$$

where $\phi$ is an exponential scaling function defined as $\phi(x) = \exp(-\beta(1-x))$, introduced in (Zhang et al., 2022). We then identify the most similar class in the visual memory to the current test sample based on cosine similarity:

$$\tilde{y} = \arg \max_{c \in \{1, \ldots, C\}} \left(\mathbf{M}_c^{\text{ada}\top} \mathbf{v}\right), \quad \mathbf{M}_c^{\text{ada}} = \text{Norm}\left(\sum_{m=1}^{S} \mathbf{S}_{c,m} \cdot \mathbf{m}_{c,m}\right), \tag{4}$$

where the visual memory is weighted by $\mathbf{S}$ before the cosine similarity computation, following (Zhu et al., 2023; Zhang et al., 2024d), to account for the relationship between the memory and the test image. Norm denotes $\ell_2$ normalization. Finally, we use the pseudo label $\tilde{y}$ to get the corresponding class-specific visual memory $\mathcal{M}_{\tilde{y}}$ as the retrieved class-relevant memory for the current test image.

**Prompt tuning.** In this step, we use the current test image $\mathbf{X}$ and the retrieved relevant visual memory $\mathcal{M}_{\hat{y}}$ to learn the textual prompt from a generic initialization $\mathbf{p}_{\text{init}}$. For the test image, we apply an entropy minimization loss, as in TPT (Shu et al., 2022), shown in Eq. 2, where we adopt random cropping as the data augmentation strategy. Concurrently, we incorporate a cross-entropy loss between the retrieved memory and the pseudo label to enhance the prompt learning. Starting from $\mathbf{p}_{\text{init}}$, we optimize the prompt to obtain $\mathbf{p}_{\text{ada}}$:

$$\mathbf{p}_{\text{ada}} = \mathbf{p}_{\text{init}} - \eta \cdot \nabla_{\mathbf{p}} \mathcal{L}_{\text{pt}} = \mathbf{p}_{\text{init}} - \eta \cdot \nabla_{\mathbf{p}} \left[\mathcal{H}\left(\bar{p}(\mathbf{X}_{[n]}, \mathbf{p})\right) - \log \bar{p}(y = \tilde{y} \mid \mathcal{M}_{\tilde{y}}, \mathbf{p})\right] \tag{5}$$

where $\eta$ denotes the learning rate. $\mathbf{X}_{[n]}$ denotes $n$ cropped patches of $\mathbf{X}$ selected from the full set of $N$ random crops $\mathbf{X}_{[N]}$ based on low prediction entropy (unlike TPT, where $\mathbf{X}_{[N]}$ refers to $N$ augmentations). $\bar{p}(\cdot)$ denotes the average predicted probability over patches of the test image or memorized patches. $\mathcal{H}(\cdot)$ denotes the entropy of a predicted probability distribution $p(\cdot)$ over $C$ classes, defined as $\mathcal{H}(p) = -\sum_{c=1}^{C} p(y = c) \cdot \log p(y = c)$.

**Memory update.** This step aims to update the multi-scale visual memory $\mathcal{M}$ with the most relevant patch from the current test image, based on the adapted textual prompt $\mathbf{p}_{\text{ada}}$. Specifically, we select a patch $\mathbf{X}_{i^*}$ from the $N$ randomly cropped views $\mathbf{X}_{[N]}$ according to vision-text similarity:

$$\hat{y} = \arg \max_c \bar{p}(y = c \mid \mathbf{X}_{[n]}, \mathbf{p}_{\text{ada}}), \quad i^* = \arg \min_{i \in \mathcal{I}} \mathcal{H}(p(\{\mathbf{X}_{[N]}\}_i, \mathbf{p}_{\text{ada}})), \tag{6}$$

$$\text{where} \quad \mathcal{I} = \left\{j : \arg \max_c p(y = c \mid \{\mathbf{X}_{[N]}\}_j, \mathbf{p}_{\text{ada}}) = \hat{y}\right\}. \tag{7}$$

We first obtain a confident prediction $\hat{y}$ by aggregating predictions over the selected subset $\mathbf{X}_{[n]}$ using the adapted prompt $\mathbf{p}_{\text{ada}}$. Then, from the subset $\mathcal{I}$ of patches whose predicted label matches $\hat{y}$, we select the patch $\mathbf{X}_{i^*}$ with the lowest prediction entropy. This operation acts as a safeguard against directly selecting the lowest-entropy patch from the entire set $\mathbf{X}_{[N]}$, which may otherwise include highly confident but irrelevant patches. Finally, we insert the selected patch into the corresponding memory slot $\mathcal{M}_{\hat{y}}$. If the memory is at full capacity, we remove the patch with the highest entropy among the existing entries and the current candidate.

These three steps for each test image constitute a round of mutual promotion between the tunable textual prompt and the evolving visual memory. Afterward, we obtain two predictions for the current test image: one from the optimized prompt and one from the updated memory $\mathcal{M}'$. We combine them to produce the final prediction:

$$P_{\text{final}} = P_{\text{pt}} + P_{\text{memo}} = P(\mathbf{y} \mid \mathbf{v}, \mathbf{p}_{\text{ada}}) + \text{Softmax}(\mathbf{M}'^{\text{ada}\top} \mathbf{v}), \tag{8}$$

where $P_{\text{pt}}, P_{\text{memo}} \in \mathbb{R}^C$. $P_{\text{memo}}$ is obtained via similarity-based classification, as in the memory retrieval step, and $\mathbf{M'}^{\text{ada}}$ is computed from the updated memory following Eqs. 3 and 4.

It is worth noting that we perform *only a single forward pass* of the CLIP image encoder for each test image and its patches, as the image encoder is frozen during the prompt tuning process. The encoded visual features are reused across all three steps, such as in Eqs. 4, 5, and 6. Therefore, we directly store the encoded features in the multi-scale visual memory, i.e., $\mathbf{M} \in \mathbb{R}^{C \times S \times d}$, in practice.

### 3.3 HOLISTIC VISUAL MEMORY

In downstream tasks, there are not only object recognition tasks but also holistic visual recognition tasks, such as land cover classification (Xiao et al., 2010) and scene understanding (Helber et al., 2019). These tasks require global visual descriptions, which may be overlooked when relying solely on image patches. Accordingly, we introduce a holistic visual memory that works in coordination with, and complements, the aforementioned multi-scale visual memory.

As shown in Algorithm 1, during memory retrieval, we use the test image as a query to retrieve relevant visual memory from both the multi-scale memory and the holistic memory, i.e., $\{\mathcal{M}, \mathcal{M}^{\text{hol}}\}$. Specifically, we compute the similarity-based probability distribution $\text{Softmax}(\mathbf{M}^{\text{ada}\top}\mathbf{v})$ using both types of memory and select the one with lower entropy to fetch the class-relevant visual memory. The prompt tuning step remains unchanged, except that the retrieved memory used in Eq. 5 is selected from either the multi-scale or holistic memory. During memory update, both types of memory update the same memory slot, $\mathcal{M}_{\hat{y}}$ and $\mathcal{M}_{\hat{y}}^{\text{hol}}$, as determined by the mechanisms in Eqs. 6 and 7. In addition, the holistic visual memory also contributes to the memory-based prediction

---

**Algorithm 1** Test-Time Prompt Tuning with Multi-Scale Visual Memory (M²TPT)

1: **Input:** Test samples $\{\mathbf{X}^t\}_{t=0}^T$; initial prompt $\mathbf{p}_{\text{init}}$; multi-scale memory $\mathcal{M}^{\text{mm}}$; holistic visual memory $\mathcal{M}^{\text{hol}}$; class-irrelevant memory $\mathcal{M}^{\text{irr}}$
2: **for** $t = 0$ **to** $T$ **do**
3:     Randomly crop the current test image $\mathbf{X}^t$ to $\mathbf{X}_{[N]}^t$.
        *// Memory retrieval*
4:     Use $\mathbf{X}^t$ to query $\mathcal{M}^{\text{mm}}$ and $\mathcal{M}^{\text{hol}}$ by Eqs. 3 and 4.
5:     Fetch the retrieved memory $\mathcal{M}_{\hat{y}}^{\text{mm}}$ or $\mathcal{M}_{\hat{y}}^{\text{hol}}$ based on similarity.
        *// Prompt tuning*
6:     **if** Retrieved memory is $\mathcal{M}_{\hat{y}}^{\text{mm}}$ **then**
7:         Optimize the prompt with the losses in Eqs. 5 and 11 using $\mathcal{M}_{\hat{y}}^{\text{mm}}$.
8:     **else**
9:         Optimize the prompt with the loss defined in Eq. 5 using $\mathcal{M}_{\hat{y}}^{\text{hol}}$.
10:    **end if**
       *// Memory update*
11:    Update $\mathcal{M}^{\text{mm}}$ and $\mathcal{M}^{\text{hol}}$ by Eqs. 6 and 7.
12:    Selectively update $\mathcal{M}^{\text{irr}}$ by Eq. 10.
       *// Prediction*
13:    Yield final prediction with the optimized prompt and updated visual memory by Eq. 8.
14: **end for**

---

$P_{\text{memo}}$, producing a prediction in the same way as the multi-scale memory. We then select the one with lower entropy as the final $P_{\text{memo}}$ in Eq. 8.

### 3.4 IRRELEVANCE SUPPRESSION

The memory retrieval and memory update processes operate without ground truth supervision at test time, making the memory inevitably noisy. To mitigate the adverse impact, we design an irrelevance suppression strategy: selectively retrieving and using class-relevant memory, while proactively penalizing class-irrelevant memory. Specifically, during memory retrieval, we filter out relatively irrelevant memory based on the similarity matrix $\mathbf{S}$:

$$\mathbf{M}_{\tilde{y}}^{\text{top}} = \mathbf{M}_{\tilde{y}} \left[ \text{TopK} \left( \mathbf{S}_{\tilde{y}}, \lfloor |\mathbf{M}_{\tilde{y}}| \cdot \gamma \rfloor \right) \right], \tag{9}$$

where $\text{TopK}(\cdot, k)$ returns the indices of the top $k$ elements with the highest similarity scores. $|\mathbf{M}_{\tilde{y}}|$ denotes the number of stored features in memory for class $\tilde{y}$, and $\gamma \in (0, 1]$ is the selection ratio. The filtered memory $\mathbf{M}_{\tilde{y}}^{\text{top}}$ is then used in the prompt tuning stage (see Eq. 5).

In addition, we construct a class-irrelevant memory $\mathcal{M}^{\text{irr}}$ to store previously seen, misleading visual cues from the test domain. Technically, we update this memory with high-confidence patches that are estimated to be irrelevant. Specifically, after memory update, given the memory prediction for the test image $\hat{y}_{\text{memo}} = \arg\max_c P_{\text{memo}}$, and the optimized-prompt-based predictions of patches $\hat{\mathbf{y}}^N = \arg\max_c p(y = c \mid \mathbf{X}_{[N]}, \mathbf{p}_{\text{ada}})$, $\hat{\mathbf{y}}^n = \arg\max_c p(y = c \mid \mathbf{X}_{[n]}, \mathbf{p}_{\text{ada}})$, if the memory prediction and the predictions of selected patches are consistent, i.e., $\sum_{j=1}^n \mathbb{1}\left[\hat{\mathbf{y}}_j^n = \hat{y}_{\text{memo}}\right] = n$,

we regard $\hat{y}_{\text{memo}}$ as a confident prediction. Then, the irrelevant memory is updated as:

$$i^* = \arg\min_{i \in \mathcal{I}} \mathcal{H}(p(\{\mathbf{X}_{[N]}\}_i, \mathbf{p}_{\text{ada}})), \quad \text{where} \quad \mathcal{I} = \left\{ i : \hat{\mathbf{y}}_i^N \neq \hat{y}_{\text{memo}} \right\}. \tag{10}$$

Here, we select the highest-confidence patch $\{\mathbf{X}_{[N]}\}_{i^*}$ among those that disagree with the confident prediction and store it in the memory slot $\mathbf{M}_{\hat{\mathbf{y}}_{i^*}^N}^{\text{irr}}$.

The class-irrelevant memory stores patches with pseudo "wrong" labels—i.e., patches that are confidently predicted to belong to a different class. This contradicts the task assumption. For example, in a label space of "cat" and "bird", an image labeled "cat" is not expected to contain a bird. To suppress these confident but irrelevant cues, we apply a flat-label KL loss:

$$\mathcal{L}_{\text{irr}} = \min\left(\frac{\alpha}{C}, \beta\right) \text{KL}\left(\frac{1}{C}\mathbf{1} \;\middle\|\; p(y = \tilde{y} \mid \mathcal{M}_{\tilde{y}}^{\text{irr}}, \mathbf{p})\right), \tag{11}$$

where $\alpha$ and $\beta$ are hyperparameters, and $C$ is the number of classes. This loss $\mathcal{L}_{\text{irr}}$ is incorporated into the prompt tuning objective $\mathcal{L}_{\text{pt}}$ to reduce the impact of the most conspicuous noise.

# 4 EXPERIMENTS

## 4.1 EXPERIMENTAL SETUP

**Datasets.** Following prior test-time prompt tuning works (Shu et al., 2022; Xiao et al., 2025), we evaluate our method on 15 datasets, including 10 downstream classification tasks—Flowers102 (Nilsback & Zisserman, 2008), DTD (Cimpoi et al., 2014), OxfordPets (Parkhi et al., 2012), StanfordCars (Krause et al., 2013), UCF101 (Soomro, 2012), Caltech101 (Fei-Fei et al., 2004), Food101 (Bossard et al., 2014), SUN397 (Xiao et al., 2010), FGVC-Aircraft (Maji et al., 2013), and EuroSAT (Helber et al., 2019)—and 5 out-of-distribution benchmarks: ImageNet (Deng et al., 2009), ImageNet-A (Hendrycks et al., 2021b), ImageNet-V2 (Recht et al., 2019), ImageNet-R (Hendrycks et al., 2021a), and ImageNet-S (Wang et al., 2019). We use the ImageNet validation set and adopt the same dataset splits as in TPT (Shu et al., 2022) for all others.

**Implementation details.** We use CLIP (Radford et al., 2021) with the ViT-B/16 encoder (Dosovitskiy et al., 2021) for all experiments. Each test image is optimized with a single prompt update step, initialized from "a photo of a [CLASS]." We adopt the AdamW optimizer (Loshchilov & Hutter, 2019) with a learning rate of $\eta = 0.003$. Only one-step prompt tuning is performed for each test sample. Random cropping follows scale $(0.08, 1)$ and aspect ratio $\left(\frac{3}{4}, \frac{4}{3}\right)$. We set $N = 32$ crops for downstream datasets and $N = 64$ for out-of-distribution datasets, with a selection ratio $n/N = 0.1$. The memory size is fixed at $S = 50$, and irrelevance suppression uses $\gamma = 0.5$, $\alpha = 5$, and $\beta = 0.1$. Additional hyperparameter details are provided in Appendix A.

## 4.2 COMPARISONS

We compare our test-time prompt tuning (TPT) method with recent TPT approaches in the upper parts of Tabs. 1 and 2, which do not involve hand-crafted or LLM-generated prompts. Moreover, we also compare our method with recent VLM adaptation approaches that utilize hand-crafted or LLM-generated prompts in the lower parts of Tabs. 1 and 2. For this comparison, we design a variant of our method by simply incorporating human-designed prompts used in DMN (Zhang et al., 2024d) into the memory update step. Specifically, the confident prediction $\hat{y}$ in Eq. 6 is obtained by combining predictions from the adapted prompt and the human-designed prompts.

**Comparisons on downstream classification tasks.** Tab. 1 presents results on 10 downstream fine-grained classification datasets. The upper part of the table compares prompt-tuning-based methods that learn a trainable prompt from a generic initialization. Compared to previous TPT methods, our approach achieves the highest accuracy on 7 datasets and yields an average improvement of 2.92%. Notably, M²TPT outperforms prior TPT methods by 15.94% on the EuroSAT dataset. The lower part of the table compares methods that leverage hand-crafted and LLM-generated prompts. We first observe that our method can benefit from incorporating human-designed prompts, achieving an average improvement of 2.9%. Compared to these methods, M²TPT attains the best performance on 8 out of 10 datasets and surpasses the second-best method by an average margin of 0.83%. Importantly, while prior TPT methods consistently underperform prompt-engineering-based approaches,

Table 1: **Results on 10 downstream image classification datasets.** The reported numbers are top-1 accuracy (%). Methods marked with * include training with labeled data from ImageNet. M²TPT† represents the version that incorporates hand-crafted and LLM-generated prompts.

| Method | Venue | Flower | DTD | Pets | Cars | UCF | Caltech | Food | SUN | Aircraft | EuroSAT | Average |
|---|---|---|---|---|---|---|---|---|---|---|---|---|
| CLIP (Radford et al., 2021) | - | 67.44 | 44.27 | 88.25 | 65.48 | 65.13 | 93.35 | 83.65 | 62.59 | 23.37 | 42.01 | 63.55 |
| *Prompt-Tuning-Based Methods* | | | | | | | | | | | | |
| CoOp * (Zhou et al., 2022b) | IJCV22 | 68.71 | 41.92 | 89.14 | 64.51 | 66.55 | 93.70 | 85.30 | 64.15 | 18.47 | 46.39 | 63.88 |
| CoCoOp * (Zhou et al., 2022a) | CVPR22 | 71.88 | 45.73 | 90.14 | 65.32 | 68.21 | 94.43 | 86.06 | 67.36 | 22.94 | 45.37 | 65.74 |
| MaPLe * (Khattak et al., 2023a) | CVPR23 | 72.23 | 46.49 | **90.49** | 65.57 | 68.69 | 93.53 | 86.20 | 67.01 | 24.74 | 48.06 | 66.20 |
| TPT (Shu et al., 2022) | NeurIPS22 | 68.98 | 47.75 | 87.79 | 66.87 | 68.04 | 94.16 | 84.67 | 65.50 | 24.78 | 42.44 | 65.20 |
| DiffTPT (Feng et al., 2023) | ICCV23 | 70.10 | 47.00 | 88.20 | 67.01 | 68.22 | 92.49 | **87.23** | 65.74 | **25.60** | 43.13 | 65.47 |
| C-TPT (Yoon et al., 2024) | ICLR24 | 69.80 | 46.00 | 88.20 | 65.80 | 65.70 | 93.60 | 83.70 | 64.80 | 24.00 | 43.20 | 64.80 |
| DynaPrompt (Xiao et al., 2025) | ICLR25 | 69.95 | 47.96 | 88.28 | 67.65 | 68.72 | **94.32** | 85.42 | 66.32 | 24.33 | 42.28 | 65.52 |
| **M²TPT** | - | **73.65** | **50.24** | 89.48 | **68.91** | **71.42** | 93.35 | 86.63 | **68.12** | 23.46 | **59.14** | **68.44** |
| *Methods Using Hand-Crafted and LLM-Generated Prompts* | | | | | | | | | | | | |
| VisDesc (Menon & Vondrick, 2022) | ICLR23 | 70.85 | 44.98 | 88.85 | 64.08 | 67.12 | 94.60 | 85.05 | 67.99 | 24.30 | 54.84 | 66.27 |
| WaffleCLIP (Roth et al., 2023) | ICCV23 | 72.35 | 45.21 | 89.95 | 63.57 | 67.19 | 94.02 | 86.68 | 67.23 | 25.39 | 55.07 | 66.67 |
| CuPL (Pratt et al., 2023) | ICCV23 | 71.30 | 44.56 | 89.13 | 65.29 | 66.83 | 92.98 | 86.11 | 62.59 | 24.90 | 47.84 | 65.15 |
| TDA (Karmanov et al., 2024) | CVPR24 | 71.42 | 47.40 | 88.63 | 67.28 | 70.66 | 94.24 | 86.14 | 67.62 | 23.91 | 58.00 | 67.53 |
| DMN (Zhang et al., 2024d) | CVPR24 | 74.49 | **55.85** | 92.04 | 67.96 | 72.51 | 95.38 | 85.08 | 70.18 | 30.03 | 59.43 | 70.40 |
| MTA (Zanella & Ben Ayed, 2024) | CVPR24 | 68.06 | 45.90 | 88.24 | 68.47 | 68.69 | 94.21 | 85.00 | 66.67 | 25.20 | 45.36 | 65.58 |
| ZERO (Farina et al., 2024) | NeurIPS24 | 67.17 | 45.86 | 87.83 | 68.97 | 69.18 | 94.41 | 86.77 | 67.63 | 25.21 | 42.17 | 65.52 |
| DPE (Zhang et al., 2024a) | NeurIPS24 | 75.07 | 54.20 | 91.14 | 67.31 | 70.44 | 94.81 | 86.17 | 70.07 | 28.95 | 55.79 | 69.40 |
| AWT (Zhu et al., 2024) | NeurIPS24 | 75.07 | 55.56 | 92.53 | 69.93 | 72.51 | **95.54** | 85.54 | 70.58 | 29.22 | 58.61 | 70.51 |
| GS-Bias (Huang et al., 2025) | ICML2025 | 71.94 | 46.10 | 90.38 | 67.33 | 67.59 | 94.60 | 86.09 | 67.40 | 26.49 | 52.42 | 67.03 |
| **M²TPT†** | - | **76.90** | 55.32 | 92.31 | 69.32 | 74.25 | 94.24 | 86.42 | 70.65 | 30.48 | 62.32 | 71.34 |

Table 2: **Results on out-of-distribution benchmark datasets.** The marked M²TPT† represents the version that incorporates hand-crafted and LLM-generated prompts.

| Method | Venue | ImageNet | ImageNet-A | ImageNet-V2 | ImageNet-R | ImageNet-S | OOD Average | Average |
|---|---|---|---|---|---|---|---|---|
| CLIP (Radford et al., 2021) | - | 66.73 | 47.87 | 60.86 | 73.98 | 46.09 | 57.20 | 59.11 |
| *Prompt-Tuning-Based Methods* | | | | | | | | |
| TPT (Shu et al., 2022) | NeurIPS22 | 68.98 | 54.77 | 63.45 | 77.06 | 47.94 | 60.80 | 62.44 |
| DiffTPT (Feng et al., 2023) | ICCV23 | 70.30 | 55.68 | **65.10** | 75.00 | 46.80 | 60.64 | 62.58 |
| C-TPT (Yoon et al., 2024) | ICLR24 | 69.30 | 52.90 | 63.40 | 78.00 | 48.50 | 60.70 | 62.42 |
| DynaPrompt (Xiao et al., 2025) | ICLR25 | 69.61 | 56.17 | 64.67 | **78.17** | 48.22 | 61.81 | 63.37 |
| **M²TPT** | - | **71.49** | **60.11** | 64.82 | 76.79 | **50.79** | **63.13** | **64.80** |
| *Methods Using Hand-Crafted and LLM-Generated Prompts* | | | | | | | | |
| VisDesc (Menon & Vondrick, 2022) | ICLR23 | 68.55 | 49.07 | 61.80 | 75.13 | 47.97 | 58.49 | 60.50 |
| WaffleCLIP (Roth et al., 2023) | ICCV23 | 68.81 | 50.78 | 62.54 | 77.49 | 49.10 | 59.98 | 61.74 |
| CuPL (Pratt et al., 2023) | ICCV23 | - | 50.72 | 63.27 | 77.05 | 49.02 | 60.02 | - |
| TDA (Karmanov et al., 2024) | CVPR24 | 69.51 | 60.11 | 64.67 | 80.24 | 50.54 | 63.89 | 65.01 |
| DMN (Zhang et al., 2024d) | CVPR24 | 72.25 | 58.28 | 65.17 | 78.55 | **53.20** | 63.80 | 65.49 |
| MTA (Zanella & Ben Ayed, 2024) | CVPR24 | 70.08 | 58.06 | 64.24 | 78.33 | 49.61 | 62.56 | 64.06 |
| ZERO (Farina et al., 2024) | NeurIPS24 | 71.17 | **62.75** | 65.23 | 80.75 | 50.59 | **64.83** | 66.10 |
| DPE (Zhang et al., 2024a) | NeurIPS24 | 71.91 | 59.63 | 65.44 | 80.40 | 52.26 | 64.43 | 65.93 |
| AWT (Zhu et al., 2024) | NeurIPS24 | 71.32 | 60.33 | 65.15 | **80.64** | 51.60 | 64.43 | 65.81 |
| GS-Bias (Huang et al., 2025) | ICML2025 | 70.57 | 56.61 | 64.62 | 80.49 | 50.33 | 63.01 | 64.52 |
| **M²TPT†** | - | **73.01** | 62.55 | **65.86** | 77.48 | 53.03 | 64.73 | **66.39** |

M²TPT with the **plain prompt** still outperforms most of these methods. This demonstrates that M²TPT effectively addresses the bottleneck of earlier TPT methods.

**Comparisons on out-of-distribution datasets.** Tab. 2 shows results on ImageNet and four out-of-distribution datasets that exhibit distribution shifts from ImageNet. In the upper part of the table, M²TPT outperforms recent test-time prompt tuning methods with an average improvement of 1.43% across the five datasets. As shown in the lower part, M²TPT also achieves state-of-the-art performance among VLM adaptation methods that leverage hand-crafted and LLM-generated prompts.

### 4.3 ABLATION STUDIES

**Ablation on main components.** We study the effectiveness of the three components in M²TPT—multi-scale visual memory, holistic visual memory, and irrelevance suppression—introduced in Secs. 3.2, 3.3, and 3.4, respectively, across 10 downstream classification datasets. We begin with a baseline that performs prompt tuning alone using selected low-entropy patches, as shown in Fig. 3a. Adding multi-scale visual memory to the baseline establishes the core framework of our method and improves the average accuracy to 67.44%, yielding a 2.44% gain. Next, we incorporate holistic visual memory, which preserves global visual context for tasks that require holistic visual understanding, resulting in a further 0.54% improvement. Finally, we intro-

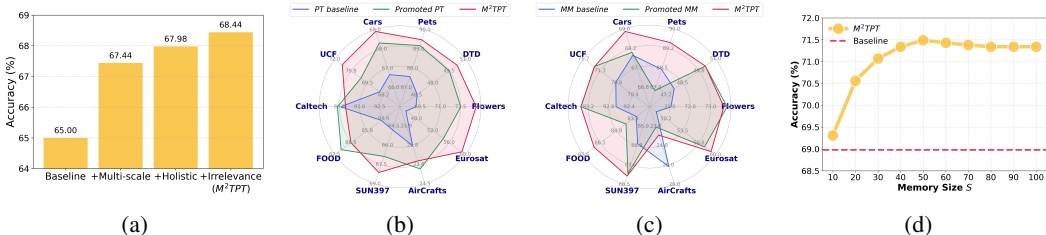

(a)  (b)  (c)  (d)

Figure 3: **(a) Ablation on main components.** Starting from a TPT baseline with low-entropy patches, we incrementally add our three components and report performance gains. **(b), (c) Analysis of the mutual promotion between the learnable prompt and the evolving memory.** (b) compares the standard test-time prompt tuning baseline with predictions $P_{pt}$ obtained from the prompt learned with visual memory. (c) compares the memory-based predictions $P_{memo}$ with a baseline where memory is updated using a static plain prompt. Improvements over the baselines demonstrate the mutual promotion between the learnable prompt and the visual memory. **(d) Effect of memory size.**

duce the irrelevance suppression strategy to better exploit the noisy test-time memory, increasing the accuracy from 67.98% to 68.44%.

**Analysis of the mutual promotion between the learnable prompt and the evolving memory.** In $M^2$TPT, visual memory provides class-relevant descriptions that enhance prompt learning, while the learned prompt in turn improves memory updates. To verify this mutual promotion, we design two baselines: prompt tuning (PT) and memory (MM). The PT baseline corresponds to standard prompt tuning with low-entropy patches. As shown in Fig. 3b, Promoted PT—i.e., $P_{pt}$ from prompts enhanced by memory—consistently outperforms PT, highlighting the benefit of multi-scale visual memory. In Fig. 3c, the MM baseline updates memory with a generic prompt, while Promoted MM—i.e., $P_{memo}$ from memory updated with learned prompts—achieves better accuracy on 8/10 datasets, showing the reverse effect. Finally, $M^2$TPT surpasses both Promoted PT and Promoted MM, demonstrating the effectiveness of jointly combining the two predictions as in Eq. 8.

**Effect of memory size.** We study the effect of memory size $S$ on the validation set of the ImageNet dataset. As shown in Fig. 3d, the Baseline refers to test-time prompt tuning without memory. $M^2$TPT shows increasing accuracy as $S$ increases from 10 to 50, consistently outperforming the Baseline. When the memory size exceeds 50, the accuracy saturates and slightly decreases.

**Computation comparison with TPT methods.** We compare the GPU memory usage and runtime of $M^2$TPT with recent TPT methods, as shown in the lower part of Tab. 3. From DTD to ImageNet, the computational cost of all TPT methods increases with the number of classes, primarily due to the higher cost of backpropagation through the text encoder. Compared to TPT (Shu et al., 2022), DynaPrompt (Xiao et al., 2025) incurs substantially higher computational burden because it optimizes multiple prompts simultaneously. By contrast, $M^2$TPT achieves significant performance improvements with only moderate additional overhead, e.g., just a 9% increase in memory usage compared to TPT.

**Discussion on TPT vs. Efficient TTA.** Beyond TPT methods, efficient TTA has recently emerged as another line of efficiency-focused approaches for VLM adaptation. These methods sidestep the inherent efficiency limitations of TPT, namely the requirement for backpropagation or double inference through the text encoder. As reported in Tab. 3, these methods are generally more efficient than TPT, and they also achieve better performance in recent studies, as summarized in Tabs. 1 and 2. Does this mean that TPT has failed completely? We argue that TPT remains valuable and merits continued exploration for two key reasons. **First**, as evidenced in Fig. 1a, Tab. 3, and Fig. 4, much of the performance gain

Table 3: **Computation resources.** The methods are evaluated on DTD and ImageNet datasets using the **plain prompt** on an H100 GPU.

| Method | DTD (47 classes) | | | ImageNet (1000 classes) | | |
|---|---|---|---|---|---|---|
| | Memory | Runtime | Acc | Memory | Runtime | Acc |
| *Efficient Test-Time Adaptation* | | | | | | |
| GS-Bias | 0.40 G | 0.02 s | 45.10 | 0.89 G | 0.03 s | 69.02 |
| MTA | 1.19 G | 0.02 s | 45.59 | 3.21 G | 0.07 s | 69.29 |
| DMN | 1.11 G | 0.03 s | 47.58 | 2.11 G | 0.04 s | 69.92 |
| *Test-Time Prompt Tuning* | | | | | | |
| TPT | 1.50 G | 0.04 s | 47.75 | 17.35 G | 0.21 s | 68.98 |
| DynaPrompt | 9.98 G | 0.41 s | 47.96 | >80G | - | 69.61 |
| **$M^2$TPT** | 1.64 G | 0.03 s | 50.24 | 18.96 G | 0.30 s | 71.49 |

Figure 4: **Average accuracy on 15 datasets.** CLIP: prompts from the CLIP authors; DMN: prompts used in DMN.

of prior efficient TTA methods comes from leveraging *hand-crafted or LLM-generated prompts*. These prompts have been carefully designed for benchmark datasets across multiple research works. However, for new real-world data, such prompt engineering (PE) lacks automation and scalability: it requires manual effort for each test scenario and assumes prior knowledge of the test data (Zhou et al., 2022b; Shu et al., 2022). In contrast, TPT methods offer on-the-fly adaptation to unseen test data streams, demonstrating flexibility and reducing reliance on human intervention. **Second**, as shown in Fig. 4, $M^2$TPT without PE achieves performance comparable to other methods that rely on it. This suggests that the prompts learned directly from test data streams in $M^2$TPT can match the effectiveness of powerful LLM-generated prompts. Thus, $M^2$TPT addresses a central bottleneck of prior TPT approaches and highlights a promising direction for future exploration in TPT.

## 5 CONCLUSION

In this paper, we identify a core limitation of prior TPT methods: they learn prompts only from the limited visual information in the current test image, making the learned prompts less competitive than those from prompt-engineering approaches. To address this, we propose test-time prompt tuning with multi-scale visual memory, enabling prompts to be learned from both accumulated class-relevant visual descriptions and the current test image. Extensive experiments show that $M^2$TPT outperforms existing TPT approaches. Moreover, by incorporating human-designed prompts, our framework further benefits from prompt engineering and achieves state-of-the-art performance compared to recent prompt-engineering-based TTA methods.

## REPRODUCIBILITY STATEMENT

We describe the $M^2$TPT framework and its components in Sec. 3, and provide implementation details, datasets, and hyperparameters in Sec. 4.1, with further details in Appendix A. All datasets used are publicly available with standard splits. To facilitate reproducibility, we will release the source code and instructions upon publication.

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

# Appendix of "Echoes of the Visual Past: Test-Time Prompt Tuning with Multi-Scale Visual Memory"

## A  MORE HYPERPARAMETER DETAILS

Following TDA (Karmanov et al., 2024), we select hyperparameters using the ImageNet validation set and **fix them across all datasets**. For $\beta$, we choose a relatively small value to maintain a low weight for the irrelevant suppression loss when the number of classes is small. We also include a sensitivity analysis on 10 downstream image classification datasets, shown in Tab. 4. The performance varies only slightly, indicating that $M^2$TPT is not highly sensitive to $\beta$.

Table 4: Sensitivity analysis for hyperparameter $\beta$ on 10 downstream image classification datasets.

|         | $\beta = 0.05$ | $\beta = 0.10$ | $\beta = 0.15$ | $\beta = 0.20$ |
|---------|---------|---------|---------|---------|
| Avg acc | 68.46 | 68.44 | 68.26 | 68.23 |

## B  ADDITIONAL COMPARISON EXPERIMENTS

We scale our evaluation from ViT-B/16 to ViT-L/14 using the plain prompt and unchanged hyperparameters. As shown in Tab. 5, $M^2$TPT achieves a 2.77% average improvement over other methods across 10 datasets, indicating robust performance gains across model scales.

Table 5: **Results with ViT-L CLIP.** The reported numbers are top-1 accuracy (%).

| Method | Flower | DTD | Pets | Cars | UCF | Caltech | Food | SUN | Aircraft | EuroSAT | Average |
|--------|--------|-----|------|------|-----|---------|------|-----|----------|---------|---------|
| TPT (Shu et al., 2022) | 76.49 | 53.55 | 93.57 | 77.96 | 75.10 | **95.21** | 89.43 | 69.40 | 31.68 | 47.56 | 71.00 |
| ZERO (Farina et al., 2024) | 76.41 | 53.63 | 94.08 | 78.39 | 74.68 | **95.21** | 90.66 | 69.61 | **33.62** | 44.21 | 71.05 |
| **$M^2$TPT** | **77.43** | **56.91** | **94.79** | **79.23** | **79.57** | 94.28 | **91.71** | **71.04** | 32.01 | **61.20** | **73.82** |

## C  ADDITIONAL ABLATION STUDIES

We compare a baseline, named Multi-scale Patch, in Tab. 6. This baseline uses multi-scale patches from the current test image for prompt tuning. The results show that our framework achieves notable performance improvements over this baseline, confirming the effectiveness of incorporating historical visual memory.

Table 6: **Baseline using multi-scale patches of the current image.** The reported numbers are top-1 accuracy (%).

| Method | Flower | DTD | Pets | Cars | UCF | Caltech | Food | SUN | Aircraft | EuroSAT | Average |
|--------|--------|-----|------|------|-----|---------|------|-----|----------|---------|---------|
| Multi-scale Patch | 69.18 | 46.57 | 87.57 | 66.67 | 68.57 | 93.47 | 85.00 | 64.80 | 22.77 | 45.44 | 65.00 |
| **$M^2$TPT** | **73.65** | **50.24** | **89.48** | **68.91** | **71.42** | 93.35 | **86.63** | **68.12** | **23.46** | **59.14** | **68.44** |

## D  ERROR BAR ANALYSIS

We conduct three runs with different random seeds, each resulting in a distinct data sample order, across 10 downstream image classification datasets. We analyze the error bars for the predictions from the adapted prompt $P_{pt}$, the visual memory $P_{memo}$, and the combined final prediction $P_{final}$ as defined in Eq. 8. As shown in Fig. 5, both $P_{pt}$ and $P_{memo}$ exhibit relatively low variance across runs, while the combined prediction shows a slightly higher standard deviation.

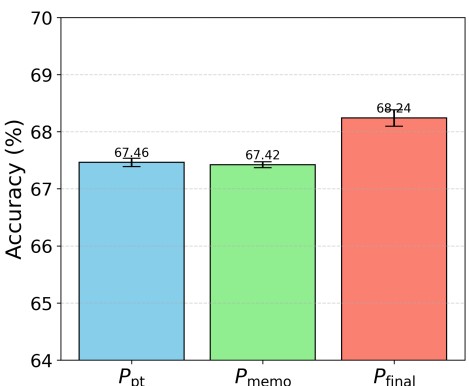

Figure 5: Results across 3 runs on 10 downstream classification datasets.

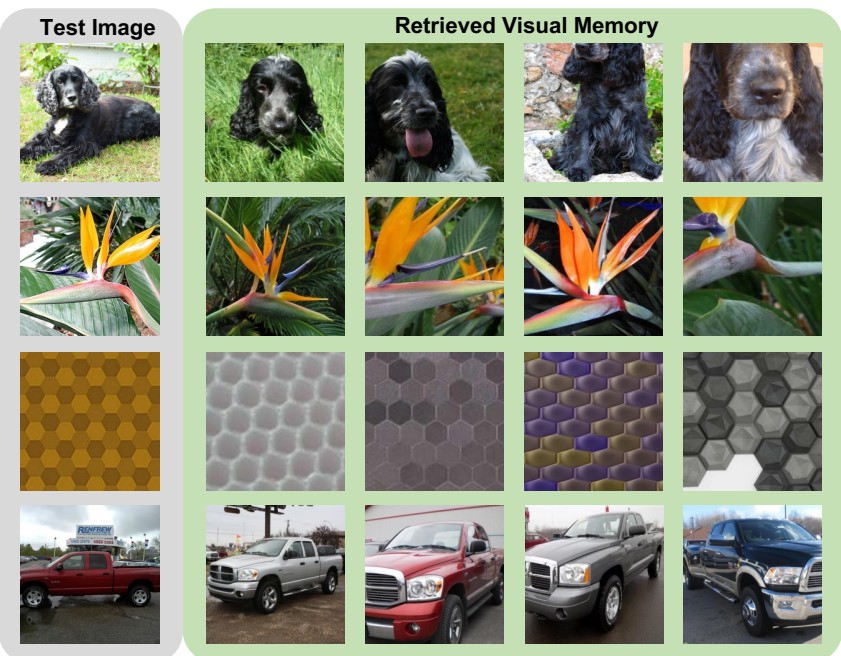

Figure 6: Visualization of visual memory retrieval.

## E  ADDITIONAL FIGURES

We present a visualization of visual memory retrieval in Fig. 6. The retrieved visual memory provides relevant visual descriptions drawn from the test distribution.

## F  BROADER IMPACTS

This work contributes to the development of adaptive vision-language systems that are more responsive to deployment-time data without requiring retraining or manual prompt engineering. By leveraging past visual information, our method enables more scalable adaptation in real-world scenarios where prior knowledge of test data is unavailable. Moreover, by reducing dependence on human-crafted or LLM-generated prompts, our approach lowers the barrier to deploying vision-language models in new domains, benefiting users without expertise in prompt engineering. However, deploying such memory-augmented models in safety-critical domains should involve strict monitoring and fail-safes to prevent over-reliance on potentially noisy or outdated information.

## G  LIMITATIONS

A primary limitation of the proposed framework—shared with other prompt-tuning-based and prompt-engineering-based adaptation methods for CLIP—is the assumption of a stable class vocabulary, which requires that all class candidates are known in advance. In settings where new classes appear dynamically, the memory structure and retrieval strategy may require further adaptation. Another limitation is that memory retrieval and update depend on the quality of pseudo-labels. A high volume of incorrect early predictions can lead to suboptimal memory updates, introducing significant noise into the adaptation process.

## H  LLM USAGE

We used large language models (LLMs) to polish the writing and improve clarity of exposition in parts of this paper. All conceptual contributions, technical content, experimental design, and analyses were conducted entirely by the authors.