# OpenReview forum: "Echoes of the Visual Past: Test-Time Prompt Tuning with Multi-Scale Visual Memory"
_ICLR.cc/2026/Conference — Submitted to ICLR 2026_

### Official Review · Reviewer_r18P · 2025-10-24

**Soundness:** 3
**Presentation:** 3
**Contribution:** 3
**Rating:** 6
**Confidence:** 4

**Summary:**

This work extends test-time prompt tuning (TPT) for vision-language models by introducing a multi-scale visual memory mechanism that stores class-relevant patch features from past test samples and uses them to guide prompt adaptation. Experiments on 15 datasets show consistent gains over previous TPT and competitive performance with hand-crafted/LLM prompts.

**Strengths:**

- It pinpoints a clear weakness in existing TPT methods, relying on a single image, and offers a conceptually reasonable solution via memory augmentation.
- Extensive experiments on 15 benchmarks, both in-distribution and OOD, with clear ablation.

**Weaknesses:**

- With regard to multi-scale visual memory, what does the term "multi-scale" refer to, and how is it initialized?
- The idea of maintaining a memory of past features (e.g., HisTPT, DynaPrompt) is not new. The main difference here lies in multi-scale patch granularity and explicit cross-promotion between memory and prompt, which is incremental rather than conceptually ground-breaking. It's better to compare with HisTPT and DYnaPrompt if possible.
- Since the memory is class-relevant, it costs 18.96G for ImageNet, so the scalability and memory efficiency are limited.
- Since only two of the compared methods were published in 2024 or 2025, are there any other recent related works?
-  Visualization of retrieved patches and how they influence prompt tokens would strengthen the paper’s interpretability.

**Questions:**

See the comments in weaknesses.

---

> ### Author Response · Authors · 2025-11-24
> **Author Response (Part 1/2)**
>
> We thank the reviewer for the constructive comments. We address all concerns and questions one by one below.
>
> > W1. With regard to multi-scale visual memory, what does the term "multi-scale" refer to, and how is it initialized?
>
>
> “Multi-scale” refers to multiple views of an image at different spatial resolutions.
> The visual memory in $M^{2}TPT$ is initially empty and gradually accumulates visual knowledge from the test stream. The memory has a fixed capacity; when full, newly added confident patches replace low-confidence ones, as described in the “Memory update” part of Sec. 3.2.
>
> > W2. The idea of maintaining a memory of past features (e.g., HisTPT, DynaPrompt) is not new. The main difference here lies in multi-scale patch granularity and explicit cross-promotion between memory and prompt, which is incremental rather than conceptually ground-breaking. It's better to compare with HisTPT and DYnaPrompt if possible.
>
>
> We discussed these two related works in the Related Work section of the initial submission. Here, we provide a more concrete comparison.
>
> **Setting.**
> DynaPrompt and HisTPT focus on a continual test-time prompt tuning setup, where the prompt is continuously optimized throughout the test stream. $M^{2}TPT$ follows the standard TPT setting introduced in TPT [a], where the prompt is adapted for each test sample.
>
> **Motivation.**
> DynaPrompt and HisTPT focus on the forgetting problem arising in continual prompt tuning, such as prompt collapse. In contrast, our work is motivated by an empirical observation that TPT methods have recently fallen behind prompt-engineering-based TTA methods. We identify a key limitation of previous TPT methods—limited visual information from a single test image for prompt learning—and $M^{2}TPT$ is designed to address this limitation.
>
> **Method Design.**
> To address the forgetting issue, HisTPT employs memory banks that store **historical text embeddings** produced by the text encoder, and uses these past outputs as a regularization signal for continuous prompt optimization. DynaPrompt instead maintains a prompt-pool mechanism containing multiple **historical prompts**, which are adaptively optimized for stable continual learning.
>
> By contrast, our method focuses on the limited visual information problem. $M^{2}TPT$ maintains multi-scale **visual memory** to provide diverse historical visual descriptions that directly enhance prompt learning, rather than regularizing with past model outputs.
>
> **Perforamnce Comparison.**
> Results for DynaPrompt are already presented in Tabs. 1 and 2 of the initial submission. For HisTPT, the official code for image classification is not available, so we copy the downstream results directly from its paper for comparison. As shown below, $M^{2}TPT$ shows performance advantages.
>
> | Method    | Flowers102 |  DTD  | Pets  | Cars  | UCF101 | Caltech101 | Food101 | SUN397 | Aircraft | EuroSAT | Average |
> |-----------|----------|-------|-------|-------|--------|---------|---------|--------|----------|---------|---------|
> | HisTPT  | 71.2 | 48.9 | 89.1 | **69.2** | 70.1 | **94.5** | **89.3** | 67.2 | **26.9** | 49.7 | 67.6|
> | DynaPrompt    | 69.95 | 47.96 | 88.28 | 67.65 | 68.72 | 94.32 | 85.42 | 66.32 | 24.33 | 42.28 | 65.52
> |$M^{2}TPT$ | **73.65**| **50.24** | **89.48** | 68.91 | **71.42** | 93.35 | 86.63 | **68.12** | 23.46 | **59.14** | **68.44**|
>
>
>
> > W3. Since the memory is class-relevant, it costs 18.96G for ImageNet, so the scalability and memory efficiency are limited.
>
> We clarify that the memory module in $M^{2}TPT$ accounts for only a small portion of the total cost—1.44 GB out of 18.96 GB, as shown in the table below.
> As reported in Tab. 3, TPT already requires 17.35 GB. Therefore, the cost growth with the number of classes mainly comes from the backpropagation required by prompt tuning, which is inherent to all TPT methods. We further discuss the efficiency point of TPT methods in the last paragraph of Sec. 4.3.
>
> ||Additional Memory (GB) |Additional Runtime (s)|
> |---|---|---|
> |DTD (47 classes)|0.13|0.01|
> |ImageNet (1000 classes)|1.44|0.09|
>
>
> > W4. Since only two of the compared methods were published in 2024 or 2025, are there any other recent related works?
>
> To our knowledge, in the past two years there have indeed been fewer new TPT methods compared with prompt-engineering–based TTA methods. A likely reason is that the latter achieve higher numbers in benchmark tables, which naturally draws more attention from the community. This observation motivated the question we raise in the Introduction — “Has the potential of TPT methods truly been exhausted?” We therefore revisited TPT from a fresh perspective, identified a core limitation in previous TPT methods, and proposed $M^{2}TPT$ to address it.

---

> ### Author Response · Authors · 2025-11-24
> **Author Response (Part 2/2)**
>
> > W5. Visualization of retrieved patches and how they influence prompt tokens would strengthen the paper’s interpretability.
>
> We have added a visualization study for visual memory retrieval in the revision. As shown in Fig. 6 in Appendix E, the retrieved visual memory provides relevant visual descriptions that come from the test distribution.
>
> We also analyze the influence of retrieved patches on prompt learning in Sec. 4.3. As shown in Fig. 3(b), prompt tuning with retrieved visual memory (green) consistently outperforms baseline prompt tuning with only the test image (blue), demonstrating the positive effect on prompt optimization.
>
> **Reference**
>
> [a] Test-time prompt tuning for zero-shot generalization in vision-language models. NeurIPS2022.

---

> > ### Comment · Reviewer_r18P · 2025-11-26
> >
> > I appreciate the authors’ rebuttal. However, I still have a few questions for discussion:
> >
> > - Regarding the multi-scale design, which specific image resolutions are used in your method, and how do they affect performance? In addition, the paper does not clearly explain that “multi-scale” corresponds to using different input resolutions.
> >
> > - At present, the memory stores features of real images. Have you considered learning a set of prototype features instead (or in addition), and if so, how might this affect performance or interpretability?

---

> > > ### Author Response · Authors · 2025-11-26
> > >
> > > Thank you for the reviewer’s follow-up questions.
> > >
> > >
> > > > Regarding the multi-scale design, which specific image resolutions are used in your method, and how do they affect performance? In addition, the paper does not clearly explain that “multi-scale” corresponds to using different input resolutions.
> > >
> > > We apologize for the confusion in our previous response. To be precise, our use of the term 'multi-scale' refers to the varying spatial extent of the random crops, not to varying input tensor resolutions. As detailed in the 'Implementation Details' (Sec 4.1), we employ a random cropping strategy with a scale range of (0.08, 1.0).
> > >
> > > - Mechanism: For every test image, we generate $N$ random crops. A crop with a small scale (e.g., 0.08) captures fine-grained, local object details, while a crop with a large scale (e.g., 1.0) captures the global object view. As described in the 'Memory update' step (Sec. 3.2), we select the most confident patch to store its feature in the visual memory, allowing the memory to accumulate diverse visual descriptions across different scales.
> > > - Resolution: Regardless of the crop's original size, all patches are resized to the standard CLIP input resolution (e.g., $224 \times 224$ for ViT-B/16) before encoding.
> > >
> > > Therefore, the visual memory is 'multi-scale' because it accumulates patch features representing diverse granularity—ranging from local textures to global shapes—rather than features extracted from images of different pixel resolutions.
> > >
> > >
> > > > At present, the memory stores features of real images. Have you considered learning a set of prototype features instead (or in addition), and if so, how might this affect performance or interpretability?
> > >
> > > Thank you for this insightful suggestion. While learning prototypes is a valid approach, we deliberately chose to store features of real image patches for two key reasons regarding **performance** and **interpretability**:
> > >
> > > - **Performance & Granularity**: As stated in the Introduction, our goal is to enrich prompt learning with "diverse, class-specific knowledge". Prototypes typically represent the centroid or abstract summary of a class, which tends to smooth out specific variations. In contrast, by storing features of individual patches (as described in the 'Memory update' step in Sec. 3.2), our memory preserves the complex, multi-modal distribution of the class (e.g., a "dog" class containing patches of both heads and tails, or different breeds). This allows the retrieval mechanism to find the specific historical patches that are most semantically similar to the current unique test image, rather than just matching a generic class average.
> > > - **Interpretability**: Storing real features offers superior interpretability. As demonstrated in Figure 6 (Appendix E), our method allows us to visualize exactly which past visual instances (patches) were retrieved to support the current prompt learning and prediction. If we used learned prototypes, these vectors would be abstract and difficult to visualize in the image space, making the decision process more opaque.
> > >
> > > Regarding the combination: We agree that adding prototypes could serve as stable "anchors" to regularize the memory. However, we believe the primary performance gain comes from the diversity of the stored instances rather than the abstraction of a prototype.
> > >
> > > We hope this additional clarification addresses the remaining concerns. Should anything remain unclear, we would be glad to continue the discussion.

---

### Official Review · Reviewer_Lgso · 2025-10-28

**Soundness:** 3
**Presentation:** 3
**Contribution:** 2
**Rating:** 4
**Confidence:** 5

**Summary:**

This paper proposes a TPT method that differs from prior approaches relying solely on a single test image and its augmented views for prompt optimization.  By introducing multi-scale visual memory, holistic memory, and an irrelevance suppression mechanism, the method aims to achieve more effective TPT. Extensive experiments across 15 datasets demonstrate competitive performance.

**Strengths:**

- The evaluation is comprehensive, including two experimental settings across 15 diverse datasets.
- The overall writing is clear and easy to follow.

**Weaknesses:**

- The method suffer from practical inefficiency. Test-time prompt tuning introduces substantial inference latency and computational overhead due to full back-propagation and multi-step forward inference. On top of this burden, the proposed approach further maintains a memory queue, which can significantly increase the computational cost, especially in tasks with a large number of classes. I am seriously concerned about the deployability of the method in real-world scenarios.
- The novelty is limited and some strongly relevant works are missing. The idea of introducing a memory mechanism into TPT is not new, as HisTPT [1] has already explored similar concepts. BoostAdaptor [2] incorporates augmented views of test images as multi-scale information in memory, which is closely related to the “multi-scale memory” proposed here. Moreover, recent approaches such as DPE [3] and GS-Bias [4] have shown more efficient test-time learning via prototype or bias updates. In addition, compared with the latest training-free method MCP [5], the proposed approach does not show clear performance advantages.

[1] Historical Test-time Prompt Tuning for Vision Foundation Models. NIPS2024

[2] BoostAdapter: Improving Vision-Language Test-Time Adaptation via Regional Bootstrapping. NIPS2024

[3] Dual Prototype Evolving for Test-Time Generalization of Vision-Language Models. NIPS2024

[4] GS-Bias: Global-Spatial Bias Learner for Single-Image Test-Time Adaptation of Vision-Language Models. ICML 2025

[5] Multi-Cache enhanced Prototype Learning for Test-Time Generalization of Vision-Language Models. ICCV 2025

**Questions:**

- The final results in Eq (8) are not solely obtained through prompt tuning, making it difficult to determine whether the method’s effectiveness primarily stems from prompt optimization.
- The tuned prompts may potentially have negative effects.  For example, TPT has been observed to decrease performance on the Pets dataset.  Such negative effects could also compromise the quality of memory samples.
- The paper does not report how many steps of prompt tuning were used, leaving unclear the computational cost and convergence behavior of the proposed method.

---

> ### Author Response · Authors · 2025-11-24
> **Author Response (Part 1/3)**
>
> We thank the reviwer for constructive comments. We address all concerns and questions one be one below.
>
> > W1. The method suffer from practical inefficiency. Test-time prompt tuning introduces substantial inference latency and computational overhead due to full back-propagation and multi-step forward inference. On top of this burden, the proposed approach further maintains a memory queue, which can significantly increase the computational cost, especially in tasks with a large number of classes. I am seriously concerned about the deployability of the method in real-world scenarios.
>
> **Clarification on “multi-step forward inference.”**\
> We would like to clarify that $M^{2}TPT$ performs only one step of prompt tuning per test image. As stated in the last paragraph of Sec. 3.2, the encoded visual features are reused across the three stages—memory retrieval, prompt tuning, and memory update. Therefore, $M^{2}TPT$ equires **only a single forward pass** of the visual encoder for each test image. Thus, as a TPT method, $M^{2}TPT$ does not add extra computation beyond the original TPT framework (TPT [a]) on the prompt-tuning side.
>
> **Computation from the memory module.**\
> Maintaining the memory queue introduces only a small overhead, even when scaling to 1000 classes. We report the additional computation introduced solely by the memory module below:
>
> ||Additional Memory (GB) |Additional Runtime (s)|
> |---|---|---|
> |DTD (47 classes)|0.13|0.01|
> |ImageNet (1000 classes)|1.44|0.09|
>
> Compared with the total computation reported in Table 3, this overhead is minor. The majority of memory usage (18.96 GB) comes from prompt tuning, such as the 17.35 GB used by the TPT [a] baseline. In other words, the main computational cost is inherent to all TPT methods, not caused by our visual memory component.
>
> **On the broader question: why TPT at all?**\
> The reviewer’s concern naturally leads to a broader question: Do we still need the TPT line of research, given the extra backpropagation cost required by TPT compared with training-free TTA methods?
>
> We believe that these two lines of research should continue to be developed in parallel, and that one should not replace the other. We present our detailed arguments on this point in:
> - The section “Prototype and Bias Optimization based TTA Methods: DPE [3] and GS-Bias [4]” in the response to W2.
> - The final paragraph of Sec. 4.3 (“Discussion on TPT vs. Backpropagation-free TTA”) in the paper.
>
> We kindly refer the reviewer to these discussions for the full reasoning relevant to W1.
>
>
> > W2. The novelty is limited and some strongly relevant works are missing. The idea of introducing a memory mechanism into TPT is not new, as HisTPT [1] has already explored similar concepts. BoostAdaptor [2] incorporates augmented views of test images as multi-scale information in memory, which is closely related to the “multi-scale memory” proposed here. Moreover, recent approaches such as DPE [3] and GS-Bias [4] have shown more efficient test-time learning via prototype or bias updates. In addition, compared with the latest training-free method MCP [5], the proposed approach does not show clear performance advantages.
>
> Thank you for pointing out these related works. We discuss them one by one below and have added all of them into the revised manuscript.
>
> ## **HisTPT [1]**
>
> Although we mentioned HisTPT in the Related Work of the initial submission, we provide a more concrete comparison here and highlight the contributions beyond it.
>
> **Motivation.**
> In $M^{2}TPT$, the multi-scale visual memory is designed to address a key limitation of prior TPT methods: prompts are learned using only limited visual information from a single test image. This limitation still exists in HisTPT, because its prompt is also optimized only with the current image. The memory in HisTPT stores **historical text embeddings** to regularize online prompt tuning and alleviate the **forgetting problem** in its continuous TPT setting.
>
> **Method.**
> The memory mechanisms differ in modality, purpose, and framework. $M^{2}TPT$ uses multi-scale **visual memory** to provide richer and more diverse visual references for prompt optimization, and introduces a mutual-promotion mechanism between memory and prompt. HisTPT, in contrast, accumulates **text embeddings** to regularize its online optimization process.
>
>
> **Setup.**
> HisTPT focuses on a continuous test-time prompt tuning setup, where the prompt is continually optimized over a stream. $M^{2}TPT$ follows the original TPT setting introduced in TPT [a], which does not involve continuous updates.
>
> Overall, while both methods contain memory concepts, the motivation, design, and contributions of $M^{2}TPT$ remain distinct from HisTPT.

---

> ### Author Response · Authors · 2025-11-24
> **Author Response (Part 2/3)**
>
> ## **BoostAdapter [2]**
>
>
> We have added discussion of this work to the Related Work section; specifically, it is a training-free TTA method with prompt engineering, similar to TDA [b] and DMN [c].
>
> Memory mechanisms are not new in training-free TTA for vision–language models. TDA and DMN use a non-parametric memory that stores historical image features to complement CLIP’s zero-shot predictions. BoostAdapter further introduces boosting samples, i.e., augmented multi-scale views of the current test image, and combines them with the historical memory at each test step. Importantly, these boosting samples provide multi-scale information **only for the current test image** and are used **only during its own prediction**. This differs from the multi-scale visual memory in $M^{2}TPT$, where multi-scale visual information is accumulated across samples and persists throughout the test stream.
>
> BoostAdapter and $M^{2}TPT$ belong to two different lines of work: training-free TTA and TPT. BoostAdapter uses multi-scale information of the current test image to complement historical memory and improve memory-based prediction. By contrast, $M^{2}TPT$ designs multi-scale visual memory to enhance TPT by providing diverse past visual descriptions for prompt learning. Therefore, BoostAdapter and $M^{2}TPT$ fall into different categories, and their motivations and method designs are distinct.
>
>
>
> ## **Prototype and Bias Optimization based TTA Methods: DPE [3] and GS-Bias [4]**
>
> We have added these two works to the Related Work section and to the main comparison tables in the revision. Similar to training-free TTA methods that use non-parametric learning to keep the text prompt available for prompt engineering, these methods also place their optimization after the encoder backbone, so the prompt remains free for prompt engineering. Therefore, they fall under the category of prompt-engineering-based methods in all our claims and discussions.
>
> We agree that these methods are efficient—just as efficiency-focused training-free TTA methods are more efficient than TPT, because they avoid all or most backpropagation that TPT methods require. However, this does not mean that TPT has completely failed. As we argue in the last paragraph of Sec. 4.3, efficiency is not the only factor to judge practicality or deployability.
>
> As shown in the table below, GS-Bias needs prompt engineering (PE) to surpass previous TPT methods. However, prompt engineering lacks automation and scalability because it requires domain expertise and manual effort to curate prompts for each scenario, and it assumes prior knowledge about the unseen test data [a][d]. In contrast, TPT methods learn prompts from the test stream on the fly, which is more flexible and practical in deployment. Furthermore, $M^{2}TPT$ **without** PE outperforms GS-Bias **with** PE. This suggests that test-time learned prompts can compete with manually engineered ones, and supports our position that TPT remains valuable and worth further exploration.
>
>
>
>
>
> | Method    | Flowers102 |  DTD  | Pets  | Cars  | UCF101 | Caltech101 | Food101 | SUN397 | Aircraft | EuroSAT | Average |
> |-----------|----------|-------|-------|-------|--------|---------|---------|--------|----------|---------|---------|
> | TPT    | 68.98 | 47.75 | 87.79| 66.87 | 68.04 | 94.16| 84.67 | 65.50 | 24.78 |42.44 | 65.20|
> | GS-Bias without PE    | 68.86      | 45.10 | 88.58 | 66.77 | 65.74  | 94.16       | 85.67   | 64.78  | 25.30    | 43.63   | 64.86   |
> | GS-Bias with PE | 71.94      | 46.10 | **90.38** | 67.33 | 67.89  | **94.60**       | 86.09   | 67.40  | **26.49**    | 52.42   | 67.03   |
> |$M^{2}TPT$ (without PE) | **73.65**| **50.24** | 89.48 | **68.91** | **71.42** | 93.35 | **86.63** | **68.12** | 23.46 | **59.14** | **68.44**|

---

> ### Author Response · Authors · 2025-11-24
> **Author Response (Part 3/3)**
>
> ## **MCP [5], performance comparison**
>
> First, we believe this paper (posted on arXiv on August 2, 2025) falls outside the required comparison scope according to the guideline:
> >*“We consider papers contemporaneous if they are published within the last two months… authors are not required to compare their work to that paper.”*(https://iclr.cc/Conferences/2026/ReviewerGuide)
>
> That said, we carefully read the paper and official code and have included it in the revised Related Work. MCP proposes a “multi-cache enhanced prototype learning” framework for TTA, whose design incorporates a cache-based mechanism similar to TDA/DMN and a prototype optimization component related to DPE. For this reason, we categorize it within the family of VLM TTA methods that use prompt engineering. Following the reviewer’s suggestion, we compare $M^{2}TPT$ with MCP++ (full version) under both prompt-engineering and general-prompt settings. Under PE, MCP++ has a slightly higher average (+0.47%), while under general prompt, $M^{2}TPT$ is higher (+0.67%). Overall, the two methods show comparable performance.
>
> However, we consider this comparison to be unfair. MCP++ uses a **separate set of hyperparameters for each test dataset**, while $M^{2}TPT$ keeps **one fixed hyperparameter set** for all datasets. Tuning hyperparameters per dataset naturally inflates numbers but is impractical for deployment. From this perspective, we believe $M^{2}TPT$ maintains a performance advantage, since its fixed-hyperparameter version matches or surpasses MCP++ tuned specifically for each dataset.
>
>
>
> | Method  | ImageNet | ImageNet-A | ImageNet-V2 | ImageNet-R | ImageNet-S | Mean|
> |-------|-----|-----|-----|-----|-----|-----|
> |**Hand-Crafted and LLM-Generated Prompts** |
> | MCP++   | 72.64 | 59.78 | 65.77 | **81.73** | **54.39** | **66.86**|
> | $M^2TPT$ | **73.01** | **62.55** | **65.86** | 77.48 | 53.03 | 66.39|
> |**General Prompt: a photo of a [class].** |
> | MCP++   | 70.22 | 56.57 | 63.37 | **78.67** | **51.84** | 64.13|
> | $M^2TPT$ | **71.49** | **60.11** | **64.82** | 76.79 | 50.79 | **64.80** |
> |**Hyperparameters**|
> |MCP++| $\alpha_1-\alpha_3=1, 0.3, 1$  $\alpha,\beta=5.5, 8.0$  learning_rate=0.00006 | $\alpha_1-\alpha_3=1, 1, 3$  $\alpha,\beta=4.0, 3.0$  learning_rate=0.0003 | $\alpha_1-\alpha_3=1, 0.3, 0.001$ $\alpha,\beta=2.5, 1.0$  learning_rate=0.00008 | $\alpha_1-\alpha_3=1, 0.3, 1$  $\alpha,\beta=1.0, 8.0$  learning_rate=0.00008 |$\alpha_1-\alpha_3=1, 0.3, 1$  $\alpha,\beta=2.0, 3.0$  learning_rate=0.0001 |
> | $M^2TPT$ | ||Fixed hyperparameters for all |
>
> > Q1. The final results in Eq (8) are not solely obtained through prompt tuning, making it difficult to determine whether the method’s effectiveness primarily stems from prompt optimization.
>
>
> $M^{2}TPT$ adopts a prompt–memory mutual promotion framework, where the learnable prompt and the evolving visual memory reinforce each other during adaptation. The final prediction in Eq. (8) combines outputs from the prompt side ($P_{pt}$) and the memory side ($P_{memo}$).
>
> We analyze these two components in Fig. 3(bc) in the Ablation Studies. Both $P_{pt}$ and $P_{memo}$ show clear improvements due to the mutual promotion mechanism. In addition, Fig. 5 shows that $P_{pt}$ achieves slightly higher accuracy than $P_{memo}$; their close performance further supports the claim that both parts benefit from the framework.
>
> > Q2. The tuned prompts may potentially have negative effects. For example, TPT has been observed to decrease performance on the Pets dataset. Such negative effects could also compromise the quality of memory samples.
>
> TPT indeed degrades on the Pets dataset, dropping from CLIP’s 88.25 to 87.79 (Table 1). In contrast, $M^{2}TPT$ achieves 89.48 on Pets, and even the prompt-only component $P_{pt}$ reaches 89.18. This demonstrates that $M^{2}TPT$ overcomes the degradation issue observed in the original TPT and improves prompt tuning on this dataset.
>
> > Q3. The paper does not report how many steps of prompt tuning were used, leaving unclear the computational cost and convergence behavior of the proposed method.
>
> $M^{2}TPT$ uses one step of prompt tuning for each test image. We have added this detail to the Implementation Details section in the revision. The computational cost was already reported in Sec. 4.3 of the initial submission.
>
>
> **Reference**
>
> [a] Test-time prompt tuning for zero-shot generalization in vision-language models. NeurIPS2022.\
> [b] Efficient test-time adaptation of vision-language models. CVPR2024.\
> [c] Dual memory networks: A versatile adaptation approach for vision-language models. CVPR2024.\
> [d] Learning to prompt for visionlanguage models. IJCV2022.

---

> ### Comment · Reviewer_Lgso · 2025-11-27
>
> Thank you to the authors for the very careful and thorough rebuttal. I would be willing to raise my score if the following concerns can be addressed:
>
> * Clarification of the method’s conceptual positioning. The proposed approach appears to be a hybrid mechanism where memory and prompt updates jointly contribute to the final prediction. However, the paper’s title implicitly frames the method strictly within the TPT paradigm, which may mislead readers, since—in principle—TPT’s output derives solely from updated prompts. I suggest that the authors refine the method’s definition either in the main text or the title to more accurately capture this interplay.
>
> * Discussion of inherited limitations from TPT. The method retains certain inherent drawbacks of the TPT paradigm, such as full-network backpropagation and the need for double inference. These limitations should be openly discussed in the paper. In particular, the efficiency analysis in Table 3 would be more complete if lighter baselines—such as TDA or GS-Bias—were included for comparison.
>
> * An interesting point for broader discussion. The substantial memory footprint induced by the TPT paradigm significantly restricts the flexibility of TTA scenarios. In many cases, its memory cost is comparable to that of few-shot training, yet without yielding clearly superior performance or efficiency. This raises an important direction for future work: pursuing more lightweight and efficient TTA methods.

---

> > ### Author Response · Authors · 2025-11-27
> >
> > We thank the reviewer for the positive feedback and for offering a clear path to raising the score.
> >
> > > Clarification of the method’s conceptual positioning. The proposed approach appears to be a hybrid mechanism where memory and prompt updates jointly contribute to the final prediction. However, the paper’s title implicitly frames the method strictly within the TPT paradigm, which may mislead readers, since—in principle—TPT’s output derives solely from updated prompts. I suggest that the authors refine the method’s definition either in the main text or the title to more accurately capture this interplay.
> >
> > We agree that $M^2TPT$ operates as a hybrid mechanism where both the optimized prompt and the visual memory contribute to the final prediction. Following the reviewer's suggestion, we have added a description of this hybrid mechanism to the method definition in the Introduction. As shown in lines 91-92 of the revision: "*This results in a dual-branch prediction mechanism where the final output is derived jointly from the tuned prompt and the evolving visual memory.*"
> >
> > > Discussion of inherited limitations from TPT. The method retains certain inherent drawbacks of the TPT paradigm, such as full-network backpropagation and the need for double inference. These limitations should be openly discussed in the paper. In particular, the efficiency analysis in Table 3 would be more complete if lighter baselines—such as TDA or GS-Bias—were included for comparison.
> >
> > We have explicitly described the inherited limitations of TPT in the "Discussion on TPT vs. Efficient TTA" in Sec. 4.3. As shown in lines 477-479: "*These methods sidestep the inherent efficiency limitations of TPT, namely the requirement for backpropagation or double inference through the text encoder.*" Furthermore, we have added the most recent work, GS-Bias, to Table 3 to make the comparison more comprehensive.
> >
> > > An interesting point for broader discussion. The substantial memory footprint induced by the TPT paradigm significantly restricts the flexibility of TTA scenarios. In many cases, its memory cost is comparable to that of few-shot training, yet without yielding clearly superior performance or efficiency. This raises an important direction for future work: pursuing more lightweight and efficient TTA methods.
> >
> > We thank the reviewer for highlighting this point. We agree that previous TPT methods incurred a substantial memory burden without delivering superior performance. This observation was a key motivation for us to rethink the TPT line of work. In this paper, we identify and address the performance limitation of prior TPT approaches (i.e., the lack of diverse visual cues), yielding a significant accuracy boost while maintaining the inherent advantage of autonomy (i.e., no need for prompt engineering). Although this work primarily addresses the performance gap, we emphasize in our rebuttal and paper that TPT and efficient/lightweight TTA are both valuable directions for the community. We advocate that these two lines of work should advance in parallel as complementary approaches, rather than being viewed as mutually exclusive. Ultimately, the choice depends on the end user's specific application—in scenarios where autonomy is critical, a significant accuracy improvement often justifies the additional computational cost.
> >
> >
> > We are happy to continue the discussion if the reviewer has further questions.

---

> ### Comment · Reviewer_Lgso · 2025-11-28
>
> Since the score-adjustment option is no longer available in the system, I am unable to formally update my rating. However, **I am  inclined to raise my score to 6 (marginally above the acceptance threshold. But would not mind if paper is rejected).**

---

> > ### Author Response · Authors · 2025-11-28
> >
> > We thank the reviewer for the prompt response. We are very glad to see that all concerns have been addressed, and we greatly appreciate the decision to raise the score to a positive rating.

---

### Official Review · Reviewer_Emou · 2025-10-31

**Soundness:** 3
**Presentation:** 2
**Contribution:** 3
**Rating:** 6
**Confidence:** 5

**Summary:**

This paper introduces M²TPT, a test-time prompt tuning method that enhances vision-language models by incorporating a multi-scale visual memory of past class-relevant image patches, allowing prompts to be learned from richer, accumulated visual context rather than just a single test image. By jointly optimizing prompts and updating memory in a mutual promotion loop—supplemented by a holistic memory for global context and an irrelevance suppression mechanism to filter noise—it outperforms existing test-time methods and even rivals performance of hand-crafted or LLM-generated prompt approaches, without requiring prior knowledge of test datasets.

**Strengths:**

- Introduces a novel test-time prompt tuning framework that bridges the performance gap between TPT and hand-crafted/LLM-generated prompts by incorporating multi-scale visual memory of past class-relevant patches.
- Achieves SOTA results on 15 benchmark datasets, outperforming prior TPT methods and prompt-engineering approaches even without human-designed prompts.
- Maintains computational efficiency with minimal overhead compared to existing TPT methods, making it practical for deployment without requiring backpropagation through the full VLM.
- Rigorous evaluation across in-distribution and out-of-distribution settings, with reproducible implementation and clear ablation studies.

**Weaknesses:**

- The method relies on pseudo-labels for memory update and retrieval, making it vulnerable to early prediction errors that can accumulate and degrade performance over time.
- Memory requires storage of visual features across the test stream, which may not be feasible in memory-constrained or real-time deployment scenarios.
- The approach assumes a fixed, known set of classes beforehand, limiting applicability to dynamic or open-category settings where new classes emerge online.
- The irrelevance suppression mechanism introduces additional complexity and hyperparameters (γ, α, β) with limited analysis of their robustness across diverse tasks or domains.

**Questions:**

See Weaknesses.

---

> ### Author Response · Authors · 2025-11-24
> **Author Response**
>
> We appreciate the reviewer’s constructive comments. Below, we address each concern and question one by one.
>
> > W1. The method relies on pseudo-labels for memory update and retrieval, making it vulnerable to early prediction errors that can accumulate and degrade performance over time.
>
> We discussed this issue in the Limitations section. Similar to other memory-based TTA methods, early prediction errors may lead to suboptimal memory updates, which introduce noise into the adaptation process. This is an inherent risk of any online, memory-involved adaptation and should be noted when using such methods.
>
> To empirically examine robustness to early pseudo-label noise, we conducted an error-bar analysis in Appendix D, running the model three times with different random seeds, each yielding a different online test sequence. We present those results in the table below. The variance across runs is low, and all runs outperform the SoTA TPT method DynaPrompt, indicating robustness to different orders of early test samples on benchmark datasets.
>
> |  | DynaPrompt | $M^{2}TPT$(run 1) | $M^{2}TPT$(run 2)| $M^{2}TPT$(run 3)|
> |--------|-----------|-----------|-----------|-------|
> |Accuracy (%) | 65.52 | 68.44 | 68.16 | 68.27|
>
>
> > W2. Memory requires storage of visual features across the test stream, which may not be feasible in memory-constrained or real-time deployment scenarios.
> fixed
>
> We would like to clarify that $M^{2}TPT$ includes a memory-management mechanism that enforces a fixed memory size of 50 entries per class, so the memory budget *does not grow with the test stream*. We report the additional computation introduced by the memory module in the table below. As shown, the extra overhead is small even for large numbers of classes.
>
> ||Additional Memory (GB) |Additional Runtime (s)|
> |---|---|---|
> |DTD (47 classes)|0.13|0.01|
> |ImageNet (1000 classes)|1.44|0.09|
>
>
>
> > W3. The approach assumes a fixed, known set of classes beforehand, limiting applicability to dynamic or open-category settings where new classes emerge online.
>
> Thank you for pointing this out. We have acknowledged such contraints in the Limitations section. The closed-set assumption is shared by all existing TPT methods and by prompt-engineering–based TTA for CLIP. Handling new, unseen categories online falls under a different research direction—out-of-distribution (OOD) detection, which aims to identify unknown classes during testing. For adapting on open-set data streams, OOD detection can be combined with TPT/TTA methods to form a complete solution. Extending $M^{2}TPT$ to such open-set settings is an interesting future work.
>
>
> > W4. The irrelevance suppression mechanism introduces additional complexity and hyperparameters (γ, α, β) with limited analysis of their robustness across diverse tasks or domains.
>
> The computation reported for the memory module in W.3 already includes the irrelevance suppression mechanism, which introduces minimal extra computational burden.
>
> Regarding hyperparameters ($\gamma$, $\alpha$, $\beta$), we would like to emphasize that all hyperparameters are **fixed across all** tasks and domains, as described in the Implementation Details. We previously included a sensitivity analysis for $\beta$ (Appendix A). We now additionally include sensitivity analyses for $\gamma$ and $\alpha$, reported below. The results on 10 downstream datasets show low variance, demonstrating the robustness of the method to these hyperparameters.
>
> |  | $\alpha=4.0$ | $\alpha=4.5$ | $\alpha=5.0$| $\alpha=5.5$|$\alpha=6.0$|
> |--------|-----------|-----------|-----------|-------|-------|
> |Accuracy (%) | 68.10 | 68.22 | 68.44 | 68.11| 68.20|
>
> |  | $\gamma=0.40$ | $\gamma=0.45$ | $\gamma=0.50$| $\gamma=0.55$|$\gamma=0.60$|
> |--------|-----------|-----------|-----------|-------|-------|
> |Accuracy (%) | 68.11 | 68.16 | 68.44 | 68.25| 67.94|

---

### Author Response · Authors · 2025-11-24
**Thanks to all the reviewers!**

We appreciate all reviewers’ positive feedback on our work, including highlighting our identification of a clear weakness in existing TPT methods [r18P], the conceptual soundness and novelty of our framework [Emou, r18P], the rigor and breadth of our evaluation [Emou, Lgso, r18P], the strong empirical performance [Emou], and the clarity of the writing [Lgso]. We also thank the reviewers for their constructive comments and thoughtful questions. We have addressed all concerns point by point in the individual responses and revised the manuscript accordingly, with changes marked in blue.

---

### Author Response · Authors · 2025-12-01
**Summary of Discussion Status and Reviewer Consensus**

Dear Area Chair,

To assist you in the evaluation process following the recent system interruption, we would like to provide a brief summary of the current discussion status.

**1. Current Status: Unanimous Positive Ratings (6, 6, 6)**
We are pleased to report that all three reviewers currently lean towards acceptance.
* **Reviewer 2 (Lgso):** Score raised **4 $\rightarrow$ 6**.
* **Reviewer 1 (Emou) & Reviewer 3 (r18P):** Current scores are **6**.

**2. Key Discussion Highlights:**
* **Addressed R2's Concerns (Score Improved):** Reviewer 2 actively engaged in the post-rebuttal discussion. After we provided additional discussion and clarifications on efficiency and related works, R2 positively acknowledged our response and explicitly raised their rating to 6.

* **Clarifications for R3:** Reviewer 3 raised constructive questions regarding term clarification and alternative technique discussions. We have provided a detailed response to these specific points in our latest comment. Although the system interruption may have prevented a final confirmation from R3, we believe our response fully resolves these technical queries and aligns with the reviewer's positive assessment.

* **R1's Support:** Reviewer 1 provided an initial positive assessment and has not raised any further concerns during the discussion period.

Thank you for your time and effort.

Sincerely,\
Authors

---

### Meta-Review · Area_Chair_zxjn · 2025-12-19

**Summary:**

The authors attribute the performance gap between test-time prompt tuning methods and handcrafted/LLM-driven prompts in VLMs to the limited visual information used during prompt learning. To address this, they equip a VLM with three types of visual memory to help learn better prompts at test time. For each test image, relevant memory items are retrieved and the text prompt is updated with a single gradient step. Empirically, the method is evaluated across multiple datasets and shows tangible improvements over several state-of-the-art methods.

Based on the discussion and the rebuttal, the AC’s reading is that all three reviewers would converge to a score of 6. The paper is generally positively received and demonstrates consistent empirical improvements across a broad set of benchmarks. However, concerns regarding novelty remain substantive. While the paper frames the method as introducing three types of memory (multi-scale, holistic, and irrelevance-related), similar memory structures and their interaction with adaptation have been studied in prior work, including historical TPT methods and memory-based adaptation frameworks inspired by the Atkinson-Shiffrin model. As a result, the proposed memory decomposition and its role in adaptation, despite being carefully engineered and empirically validated, appear somewhat incremental.

Beyond novelty, some aspects of the method are insufficiently examined. The prompt update is fixed to a single gradient descent step, and the paper does not analyze the effect of multiple update steps or the sensitivity of performance to this design choice (despite computational burden, this choice needs study). In addition, while the paper includes ablations of the memory components, the central premise, that prompt tuning underperforms primarily because it lacks sufficient visual information, is not directly validated. This makes it difficult to clearly attribute the observed gains to the proposed mechanism rather than to the general use of memory, which has been explored previously. For these reasons, despite the generally positive reception, the AC sadly recommends rejection.

**Reviewer Concerns:**

Several concerns were raised regarding the proposed method, including
- the use of memory at test time,
- the need for backpropagation through the text encoder, and
- the level of novelty over existing test-time adaptation approaches.

The rebuttal addressed some, but not all, of these concerns. In particular, the authors clarified the computational pipeline, explained that the method performs only one gradient update per test image, and better positioned the work with respect to existing TPT and training-free TTA methods. The memory size was discussed, and additional arguments were provided on why TPT remains relevant despite its higher computational cost; the AC agrees with the authors on this point. That said, some concerns remain outstanding.
- The novelty concern is not fully resolved. While the paper proposes a specific memory design and its integration with prompt tuning, the core idea of using memory for test-time adaptation has been explored in prior work.
- Even with a fixed budget, the memory overhead remains substantial (around 1–1.5 GB), which limits scalability and deployability, as acknowledged by the reviewers.
- The method also uses backpropagation at test time; however, no analysis is provided on the optimality or sensitivity of using only one gradient descent step.
- In addition, from the AC’s perspective, there is a potential mismatch between memory updates and prompt usage. Memory entries are updated based on predictions from an adapted prompt, while prompts are re-initialized for every test images. It is therefore unclear whether the retrieved memory will always remain well aligned with future test samples, and this aspect is not examined in the paper.

**Reviewer Scores:**

Based on the discussion, two reviewers (Emou and r18P) would likely have maintained their original scores of 6. Reviewer Lgso initially expressed stronger concerns and scored the paper as 4, but explicitly indicated during the discussion that they were inclined to raise their score to 6 after the authors addressed their concerns.

---

### Decision · Program_Chairs · 2026-01-26

Reject